# MPDZ promotes DLL4-induced Notch signaling during angiogenesis

**Fabian Tetzlaff[1,2], M Gordian Adam[1†], Anja Feldner[1], Iris Moll[1], Amitai Menuchin[3], Juan Rodriguez-Vita[1], David Sprinzak[3], Andreas Fischer[1,2,4]\***

[1]Division of Vascular Signaling and Cancer, German Cancer Research Center (DKFZ), Heidelberg, Germany; [2]European Center for Angioscience, Medical Faculty Mannheim, Heidelberg University, Mannheim, Germany; [3]Department of Biochemistry and Molecular Biology, Wise Faculty of Life Science, Tel Aviv University, Tel Aviv, Israel; [4]Medical Clinic I, Endocrinology and Clinical Chemistry, Heidelberg University Hospital, Heidelberg, Germany

**Abstract** Angiogenesis is coordinated by VEGF and Notch signaling. DLL4-induced Notch signaling inhibits tip cell formation and vessel branching. To ensure proper Notch signaling, receptors and ligands are clustered at adherens junctions. However, little is known about factors that control Notch activity by influencing the cellular localization of Notch ligands. Here, we show that the multiple PDZ domain protein (MPDZ) enhances Notch signaling activity. MPDZ physically interacts with the intracellular carboxyterminus of DLL1 and DLL4 and enables their interaction with the adherens junction protein Nectin-2. Inactivation of the MPDZ gene leads to impaired Notch signaling activity and increased blood vessel sprouting in cellular models and the embryonic mouse hindbrain. Tumor angiogenesis was enhanced upon endothelial-specific inactivation of MPDZ leading to an excessively branched and poorly functional vessel network resulting in tumor hypoxia. As such, we identified MPDZ as a novel modulator of Notch signaling by controlling ligand recruitment to adherens junctions.

DOI: https://doi.org/10.7554/eLife.32860.001

**\*For correspondence:**
a.fischer@dkfz.de

**Present address:** †Metanomics Health GmbH, Berlin, Germany

**Competing interests:** The authors declare that no competing interests exist.

## Introduction

Angiogenesis, the process of forming new blood vessels from pre-existing ones is essential for embryonic development, tissue growth, wound healing, and regeneration. However, angiogenesis also substantially contributes to the pathogenesis of several diseases, most notably tumor progression (*Potente et al., 2011*). Angiogenesis is stimulated by vascular endothelial growth factor (VEGF), which activates quiescent endothelial cells (EC). These cells subsequently degrade the extracellular matrix and migrate toward the VEGF gradient. A new vessel sprout is guided by the tip cell, which expresses high amounts of VEGF receptors (*Siekmann et al., 2013*). VEGF signaling induces the expression of the Notch Delta-like ligand 4 (DLL4) on activated ECs. This transmembrane protein activates Notch receptors on adjacent cells, which adopt the stalk cell phenotype and form the new vessel lumen.

Notch signaling requires ligand binding in trans that triggers Notch receptor cleavage to release the intracellular domain (NICD). NICD translocates to the nucleus where it acts as a transcriptional regulator. Prototypical Notch target genes are the HES and HEY transcriptional repressors. These inhibit VEGF receptor expression which limits responsiveness to VEGF, tip cell formation and vessel branching (*Bray, 2016*; *Fischer et al., 2004*).

Inhibition of DLL4/Notch signaling is a powerful tool to interfere with angiogenesis as it results in the formation of excessive tip cell numbers and vessel branches. This chaotic vessel network precludes proper blood perfusion leading to severe tumor hypoxia. Interestingly, inactivation of the

**eLife digest** Blood vessels transport oxygen and nutrients to all our organs and also remove waste products. New blood vessels form – in a process called angiogenesis – when a tissue is not receiving enough oxygen. This happens during normal development and wound healing, but also during tumor growth. Cells at the tip of a branching blood vessel sense when a tissue lacks oxygen and use proteins on their cell surfaces to help new vessels to grow.

During this process, the tip cells of an existing vessel relay the signal from the tissue to other cells 'behind' them, in the so-called stalk of the vessel. It is known that tip- and stalk cells communicate by using specific proteins at their interfaces. The tip cells activate proteins called Notch ligands, such as DLL4, while stalk cells express the Notch receptor. During a process called Notch signaling, the ligands bind to the receptor, which becomes active and helps to control angiogenesis. It also hinders excessive vessel branching and so prevents the blood vessels from becoming leaky and inefficient. However, it was not known exactly how Notch ligands interact with their receptors on neighboring cells, and Notch signaling is regulated.

Here, Tetzlaff et al. sought to answer these questions by using blood vessel cells from the human umbilical cord grown in the laboratory and blood vessel cells in mice. The results showed that the proteins DLL1 and DLL4 interacted with a protein called MPDZ. This interaction stabilized the DLL proteins at the cell membrane, which increased the Notch-signaling activity.

When Tetzlaff et al. experimentally reduced the amount of MPDZ in the laboratory-grown cells, the Notch signaling decreased. Furthermore, the cells with less MPDZ formed more branching structures. And when MPDZ was genetically removed in mice, the embryos had more branched blood vessels in their developing brains. Lastly, when mice without MPDZ were transplanted with tumor cells, the tumors contained more, but leakier, blood vessels and were not supplied with enough oxygen.

This suggests that MPDZ is an important factor that helps to regulate angiogenesis by enhancing Notch signaling between tip and branch cells in a new blood vessel. The increased activity of the Notch limits new blood vessels from branching too much. A better understanding of how blood vessels form or become leaky may help to find ways to prevent tumors from growing.
DOI: https://doi.org/10.7554/eLife.32860.002

Notch ligand Jagged1 (JAG1) results in a reduced sprouting angiogenesis, indicating that JAG1 and DLL4 have opposing roles during blood vessel formation (*Benedito et al., 2009*; *Kangsamaksin et al., 2015*; *Kofler et al., 2011*).

It remains poorly understood which factors modulate and control Notch activity during angiogenesis. To become fully competent for Notch receptor activation, Notch ligands need to gain several posttranslational modifications, for example ubiquitinylation (*D'Souza et al., 2008*). In addition, Notch ligands need to be presented on the cell surface at an area, that is likely to be in contact with Notch receptors on adjacent cells (*Shaya et al., 2017*). Delta-like and Jagged proteins contain different PDZ binding motifs at their intracellular carboxyterminus, which enable binding to certain PDZ domain containing proteins. There are some indications that binding of Notch ligands to PDZ domain proteins, for example DLL1 to MAGI1 and SYN2BP as well as JAG1 to AF6 control their cellular localization or their protein stability (*Adam et al., 2013*; *Ascano et al., 2003*; *Mizuhara et al., 2005*). Interestingly, these proteins are associated with either adherens or tight junctions. Notch receptors are also localized at adherens junctions in several cell types (*Batchuluun et al., 2017*; *Benhra et al., 2011*; *Hatakeyama et al., 2014*; *Sasaki et al., 2007*). Therefore, clustering ligands and receptors at cellular junctions might increase the rate of physical binding events and subsequent Notch signaling activity.

In yeast, two-hybrid screening approaches the Notch ligands DLL1 and DLL4, but not JAG1, interacted with the multiple PDZ domain protein (MPDZ) also known as MUPP1 (*Adam et al., 2013*; *Estrach et al., 2007*). Also a synthetic DLL1 peptide containing the 27 carboxyterminal amino acids interacted with PDZ proteins including MPDZ (*Wright et al., 2004*). However, this interaction had not yet been confirmed by independent methods and any potential functional consequences are elusive. MPDZ contains 13 PDZ domains and a single L27 domain (*Ullmer et al., 1998*). It lacks an

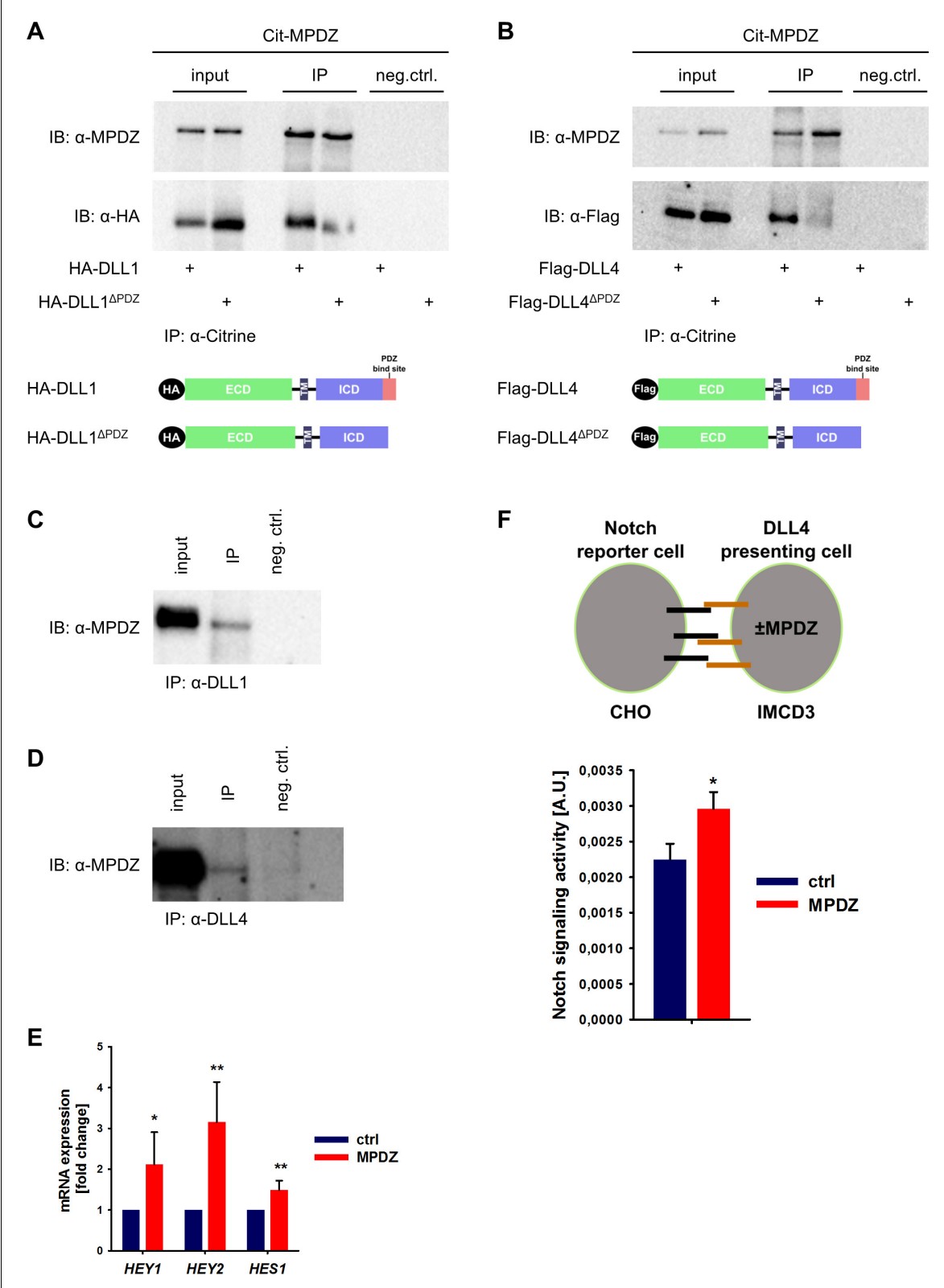

**Figure 1.** MPDZ interacts with DLL1 and DLL4. (**A, B**) HEK293T cells were transfected with Citrine-MPDZ together with HA-tagged DLL1, HA-tagged DLL1$^{\Delta PDZ}$ (lacking the PDZ-binding site), Flag-tagged DLL4 or Flag-tagged DLL4$^{\Delta PDZ}$. Antibodies against Citrine were used to immunoprecipitate Citrine-MPDZ. HA and FLAG-tagged proteins as well as MPDZ were detected by immunoblot (IB). Scheme shows structures of the constructs used for co-immunoprecipitation. Input, 10% of the immunoprecipitate. Cit-MPDZ, Citrine-MPDZ; IP, immunoprecipitation; neg.ctrl., negative control. (**C, D**)
*Figure 1 continued on next page*

*Figure 1 continued*

MPDZ was co-expressed with either DLL1 or DLL4 in primary endothelial cells (HUVEC). DLL1 and DLL4 were pulled down by using specific antibodies. DLL1, DLL4 and MPDZ were detected by immunoblot (IB). Input, 5% of the immunoprecipitate. IP, immunoprecipitation; neg.ctrl., negative control. (E) HUVEC were either transduced with adenovirus expressing GFP (ctrl) or MPDZ (MPDZ). Expression level of Notch target genes *HEY1*, *HEY2* and *HES1* were analyzed by qPCR 48 hr after transduction. Data are presented as mean ±SD. n = 4; *, p<0.05; **, p<0.01 unpaired Student's t-test. (F) Scheme of the co-culture Notch reporter assay. IMCD3 cells expressing the Notch ligand DLL4 were co-cultured with CHO-N1-CIT cells carrying a Notch luciferase reporter construct. The IMCD3 sender cells were modified by expression of MPDZ or an empty vector control. After 48 hours, cells were lysed and the light emission of the luciferin and the Renilla luciferase activities were measured. Signaling activity is calculated by normalizing the luciferase signal with the Renilla signal. Data are presented as mean ± SEM. n = 5; *, p<0.05 unpaired Student's t-test.

DOI: https://doi.org/10.7554/eLife.32860.003

The following source data and figure supplement are available for figure 1:

**Source data 1.** Source data of qantitative PCR analysis related to *Figure 1E*.
DOI: https://doi.org/10.7554/eLife.32860.005
**Figure supplement 1.** MPDZ interacts with DLL1 and DLL4.
DOI: https://doi.org/10.7554/eLife.32860.004

intrinsic catalytic function, and it is assumed that its function is to cluster proteins at the cell membrane or at adherens and tight junctions (*Adachi et al., 2009*). Such clustering of proteins was shown to affect the strength of melatonin or the AMPA transmembrane receptor signaling (*Guillaume et al., 2008*; *Krapivinsky et al., 2004*). As Notch receptor expression is often enriched at cellular junctions (*Batchuluun et al., 2017*; *Benhra et al., 2011*; *Hatakeyama et al., 2014*; *Sasaki et al., 2007*), we analyzed how the protein interaction of MPDZ with the Notch ligands DLL1 and DLL4 affects Notch signaling during angiogenesis.

## Results

### MPDZ physically interacts with the Notch ligands DLL1 and DLL4 and promotes Notch signaling

MPDZ has been identified in screening approaches as a putative binding partner of the Notch ligands DLL1 and DLL4 (*Adam et al., 2013*; *Estrach et al., 2007*; *Wright et al., 2004*). To verify this, we performed co-immunoprecipitation studies in HEK293T cells. Full-length DLL1 and DLL4 or mutants thereof lacking the carboxyterminal PDZ-binding site (amino acids IATEV) were co-expressed with MPDZ fused with an amino-terminal fluorescent Citrine. Co-immunoprecipitation revealed that MPDZ associated with DLL1 and DLL4 proteins. However, DLL1 or DLL4 lacking their PDZ-binding site did not interact with MPDZ (*Figure 1A and B*), indicating that the carboxyterminus of Delta-like ligands binds to MPDZ. This protein-protein interaction could also be detected in primary human umbilical venous endothelial cells (HUVEC) as well as in whole murine kidney lysates using a co-immunoprecipitation approach (*Figure 1C and D*, *Figure 1—figure supplement 1A and B*).

Since SYNJ2BP binds also to DLL1 and DLL4 via the PDZ-binding motif and induces Notch signaling, a competition between SYNJ2BP and MPDZ might be possible. However, pull-down studies showed that the absence of MPDZ did not overtly affect the binding of SYNJ2BP to DLL1 or DLL4 (*Figure 1—figure supplement 1C and D*).

The activity of Notch signaling depends critically on the amount of active DLL1/4 molecules on the cell surface. We tested whether the MPDZ-DLL1/4 protein interaction could alter Notch signaling activity. Forced expression of MPDZ promoted Notch signaling in HUVEC as indicated by higher expression levels of the Notch target genes *HEY1*, *HEY2* and *HES1* (*Figure 1E*). To test if MPDZ would alter the ability of DLL4 to activate Notch receptors in trans, IMCD3 cells expressing DLL4 (*sender cells*) were transfected with plasmids encoding MPDZ cDNA or empty vector control. A Notch luciferase reporter CHO cell line (*receiver cells*) was co-cultured with the IMCD3 *sender cells* (*Figure 1F*). This showed that higher amounts of MPDZ in the DLL4 expressing *sender cells* resulted in increased Notch signaling activity in *receiver cells* (*Figure 1F*).

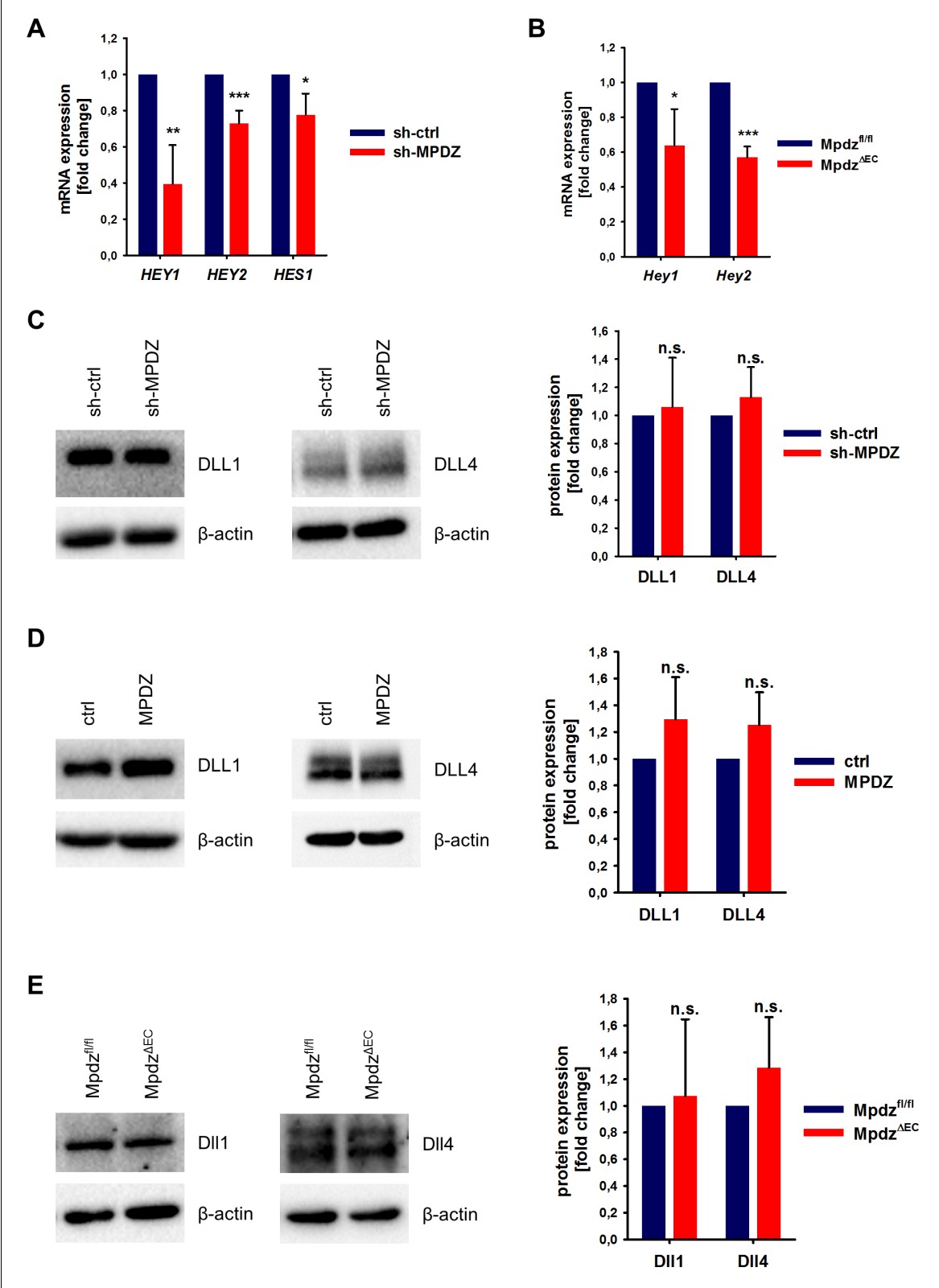

**Figure 2.** MPDZ promotes Notch signaling activity. (A) HUVECs were either transduced with lentivirus expressing GFP (sh-ctrl) or with lentivirus expressing shRNA against MPDZ (sh-MPDZ). Expression level of Notch target genes *HEY1*, *HEY2* and *HES1* were analyzed by qPCR 48 hr after transduction. Data are presented as mean ±SD. n ≥ 3; *, p<0.05; **, p<0.01; ***, p<0.001 unpaired Student's t-test. (B) Cardiac endothelial cells were isolated from *Mpdz*<sup>fl/fl</sup> and *Mpdz*<sup>ΔEC</sup> mice by magnetic beads bound with CD31 antibodies. Expression levels of Notch target genes *Hey1* and *Hey2*

*Figure 2 continued on next page*

*Figure 2 continued*

were analyzed by qPCR. Data are presented as mean ±SD. n = 3; *, p<0.05; ***, p<0.001 unpaired Student's t-test. (**C**) HUVECs were either transduced with lentivirus expressing GFP (sh-ctrl) or with lentivirus expressing shRNA against MPDZ (sh-MPDZ). Expression levels of DLL1 and DLL4 were analyzed by immunoblotting 48 hr after transduction. β-actin served as loading control. Data are presented as mean ±SD. n ≥ 3; n.s., not significant. (**D**) HUVECs were either transduced with adenovirus expressing GFP (ctrl) or with adenovirus expressing MPDZ. Expression levels of DLL1 and DLL4 were analyzed by immunoblotting 48 hr after transduction. β-actin served as loading control. Data are presented as mean ±SD. n ≥ 3; n.s., not significant. (**E**) Lung endothelial cells were isolated from *Mpdz*^fl/fl^ and *Mpdz*^ΔEC^ mice by CD31 magnetic beads. Protein amounts of Dll1 and Dll4 were analyzed by immunoblotting. β-actin served as loading control. Data are presented as mean ±SD. n = 3; n.s., not significant.

DOI: https://doi.org/10.7554/eLife.32860.006

The following source data is available for figure 2:

**Source data 1.** Source data of qantitative PCR analysis related to *Figure 2A and B*.

DOI: https://doi.org/10.7554/eLife.32860.007

## Loss of MPDZ impairs endothelial Notch signaling in vitro and in mice

To test if MPDZ contributes to basal Notch signaling in ECs, we silenced MPDZ expression in HUVEC using established lentiviruses expressing independent shRNAs (*Feldner et al., 2017*). The reduction of MPDZ expression (93 ± 3%, n = 4, p<0.001) resulted in a significant reduction of mRNA expression of the Notch target genes *HEY1*, *HEY2* and *HES1* (*Figure 2A*), indicating diminished Notch activity.

This could also be observed in cardiac CD31-positive ECs derived from EC-specific *Mpdz*-deficient mice (*Tek*-Cre;*Mpdz*^fl/fl^ referred to as *Mpdz*^ΔEC^) (*Feldner et al., 2017*). qPCR analysis revealed that the reduction of *Mpdz* expression (69 ± 12%, n = 4, p<0,001) resulted in a diminished Notch signaling activity as the relative *Hey1* and *Hey2* mRNA amounts were lower compared to Cre-negative littermate controls (*Figure 2B*).

## MPDZ promotes cell surface localization of DLL1 and DLL4

Next, we aimed at elucidating the mechanism of how MPDZ alters the activity of Notch ligands. The total expression levels of DLL1 and DLL4 proteins in HUVEC lysates were not altered upon silencing of *MPDZ* (*Figure 2C*). The same was observed after adenoviral *MPDZ* overexpression (*Figure 2D*). Lung ECs derived from *Mpdz*^ΔEC^ mice also did not show changes in total Dll1 and Dll4 protein expression levels compared to controls (*Figure 2E*).

MPDZ is able to cluster several transmembrane proteins at tight and adherens junctions. For instance, MPDZ facilitates RhoA signaling by recruiting Syx to endothelial junctions (*Ngok et al., 2012*). The activation of Notch receptors requires the interaction with Notch ligands presented on the cell surface. MPDZ is expressed in ECs derived of arteries, veins and microvessels with pronounced localization at the cell membrane (*Figure 3—figure supplement 1*). To address whether MPDZ affects the cell surface expression of the Notch ligands DLL1 and DLL4, we first tested if the carboxyterminal PDZ-binding sites of these Notch ligands are important for cell surface presentation. Constructs in which either full length DLL1 and DLL4 or versions lacking the PDZ-binding motifs were fused to a mCherry tag were generated. Those constructs were expressed in HUVEC and cell surface expression was analyzed by flow cytometry. Only mCherry-positive HUVEC were gated and the cell surface presentation of the Notch ligands was analyzed using specific antibodies against the extracellular domains of DLL1 and DLL4, respectively. Cell surface expression levels of full length DLL1 and DLL4 were higher than those of their mutants lacking the PDZ-binding site (*Figure 3A*). This indicates that protein-protein interactions via the PDZ-binding site could influence the localization of Delta-like proteins.

To test whether indeed MPDZ is involved in regulating DLL4 cell surface localization, we analyzed cell surface presentation on isolated mouse embryonic EC at the developmental stage E11.5 by flow cytometry. This revealed that the Dll4 cell surface expression of CD34-positive ECs was lower in *Mpdz*-deficient cells compared to wild type littermate controls (*Figure 3B*).

## MPDZ recruits DLL1 and DLL4 to the adherens junction protein nectin-2

MPDZ binds to the intracellular domain of the single-pass type I membrane glycoprotein Nectin-2, which is a component of adherens junctions (*Adachi et al., 2009*) and is expressed in the

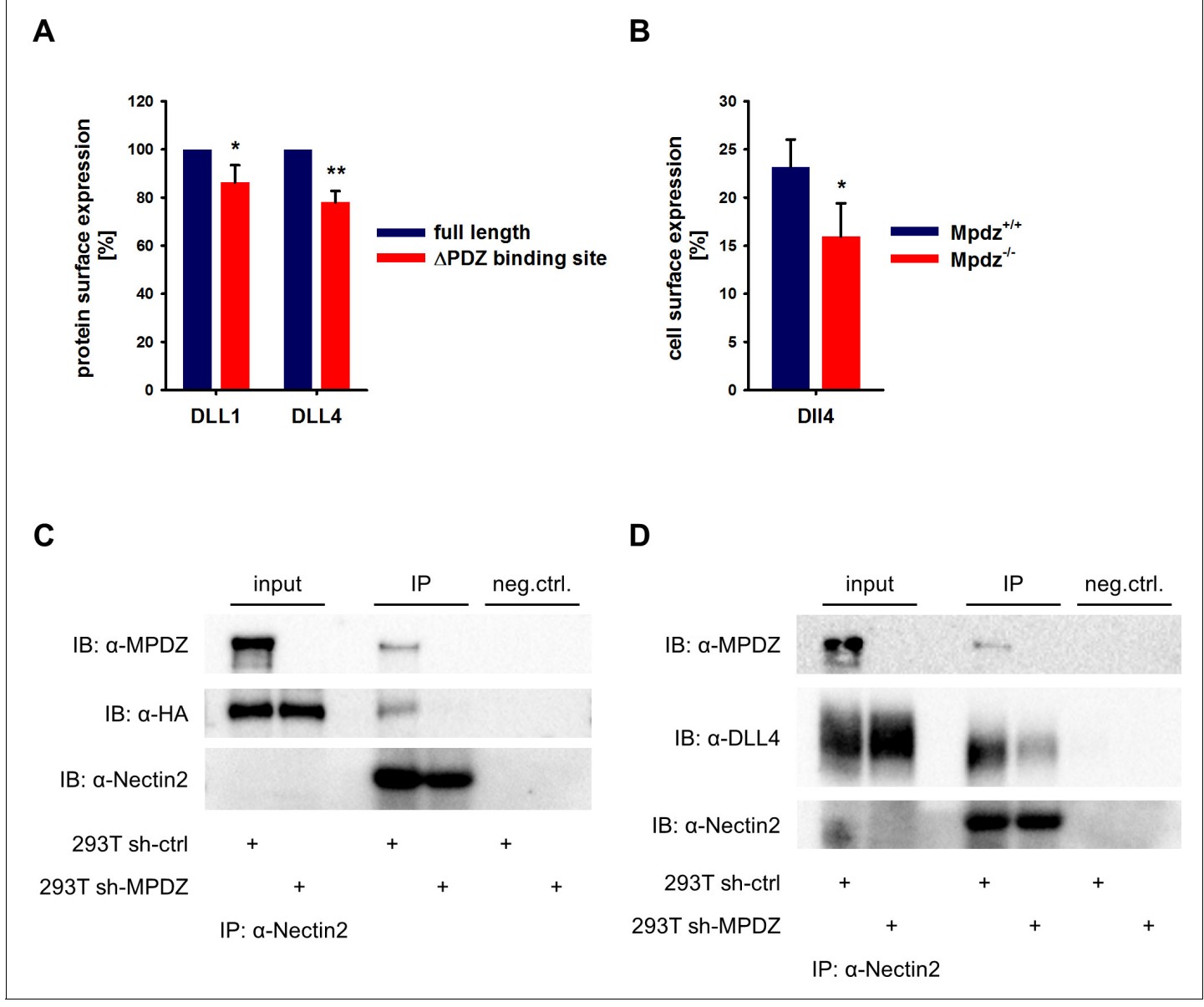

**Figure 3.** MPDZ recruits DLL1 and DLL4 to Nectin-2. (A) DLL1 and DLL4 full length and such lacking the PDZ-binding site (ΔPDZ) constructs containing a mCherry tag were expressed in HUVEC. Cells were stained with antibodies against DLL1 or DLL4. DLL1 and DLL4 surface expression of mCherry positive cells was analyzed by flow cytometry. n = 3; *, p<0.05; **, p<0.01 unpaired Student's t-test. (B) Endothelial cells were isolated from *Mpdz*$^{+/+}$ embryos and *Mpdz*$^{-/-}$ littermates at embryonic day E11.5. Cells were purified by CD31 magnetic dynabeads and stained with anti-CD34 and anti-Dll4 antibodies for flow cytometric analysis. n = 4; *, p<0.05; unpaired Student's t-test. (C, D) HEK293T control cells (293T sh-ctrl) as well as MPDZ-silenced HEK293T cells (293T sh-MPDZ) were transfected with Nectin-2 and HA-tagged DLL1 or Flag-tagged DLL4. For immunoprecipitation a Nectin-2 antibody was used and HA-tagged DLL1, Flag-tagged DLL4 and MPDZ were detected by western Blotting. Input, 10% of immunoprecipitate. IB, Immunoblot; IP, Immunoprecipitation; neg.ctrl., negative control.

DOI: https://doi.org/10.7554/eLife.32860.008

The following source data and figure supplement are available for figure 3:

**Source data 1.** Source data of FACS analysis related to *Figure 3A and B*.
DOI: https://doi.org/10.7554/eLife.32860.010
**Figure supplement 1.** MPDZ recruits DLL1 and DLL4 to Nectin-2.
DOI: https://doi.org/10.7554/eLife.32860.009

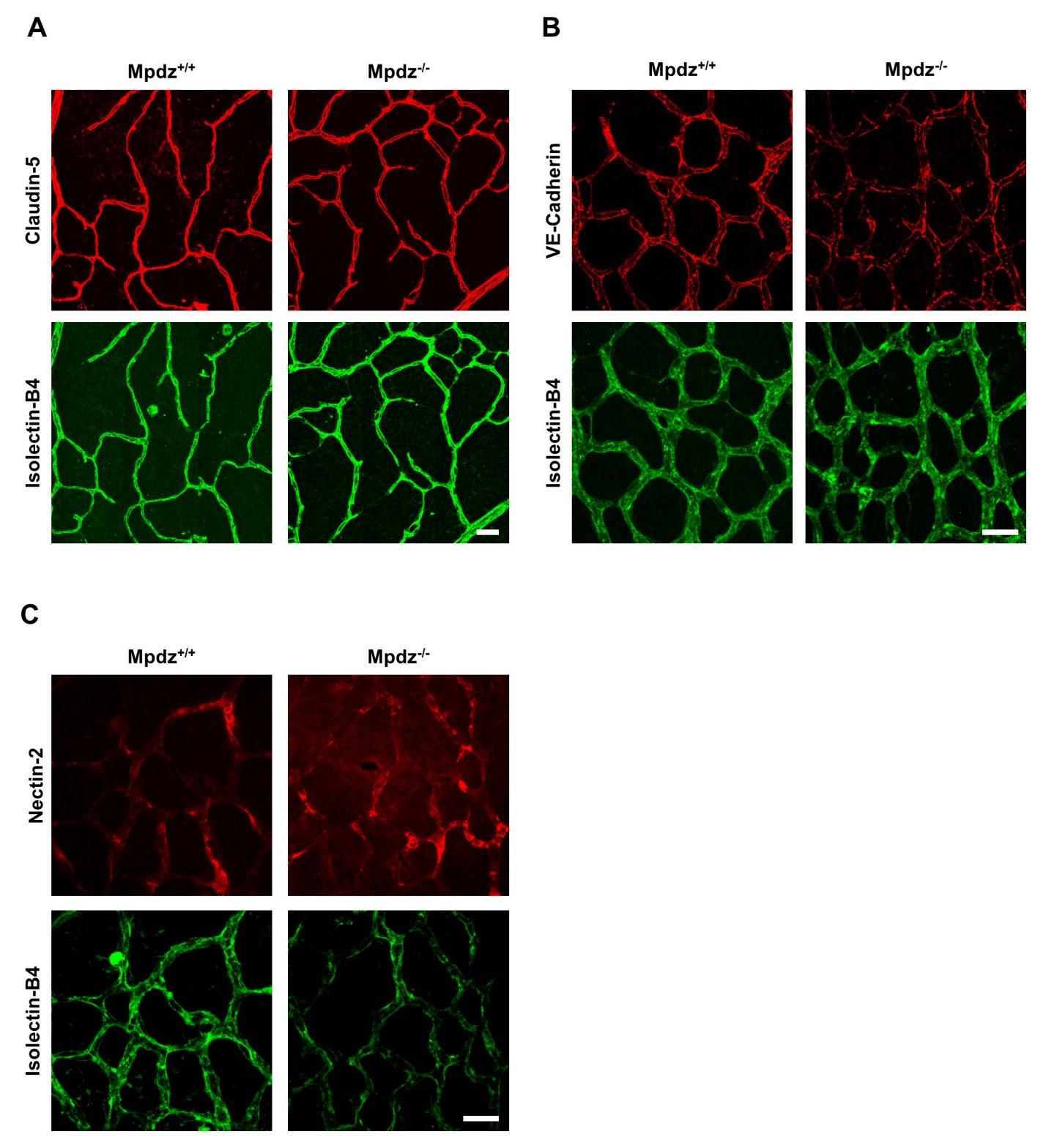

**Figure 4.** Mpdz does not affect cell cell junction assembly. (A) Retinae isolated from 14 days old *Mpdz*$^{-/-}$ pups and control littermates (*Mpdz*$^{+/+}$) were stained with Isolectin-B4 and antibodies against Claudin-5. Images were acquired with the confocal microscope LSM 700. Scale bar: 25 µm. (B, C) Retinae were isolated 9 days old *Mpdz*$^{-/-}$ pups and control littermates. Staining for endothelial cells with Isolectin-B4 and VE-Cadherin or Nectin-2. Images were acquired with the confocal microscope LSM 700. Scale bar: 25 µm.

DOI: https://doi.org/10.7554/eLife.32860.011

*Figure 4 continued on next page*

*Figure 4 continued*

The following figure supplements are available for figure 4:

**Figure supplement 1.** Mpdz does not affect cell junction assembly.
DOI: https://doi.org/10.7554/eLife.32860.012
**Figure supplement 2.** Mpdz does not affect cell cell junction assembly.
DOI: https://doi.org/10.7554/eLife.32860.013

endothelium (*Rehm et al., 2013*; *Wallez and Huber, 2008*). Thus, MPDZ could be involved in recruiting DLL1 and DLL4 to Nectin-2 containing adherens junctions. To test this, we performed co-immunoprecipitation studies in HEK293 cells. This revealed that DLL1 and DLL4 could be co-immunoprecipitated with Nectin-2. However, silencing of *MPDZ* expression abolished the interaction of DLL1 and DLL4 with Nectin-2 (*Figure 3C and D*).

Interestingly, loss of *Mpdz* in mice did not affect the overall formation of vascular tight junctions (e.g. staining patterns of Claudin-5 and Occludin were unremarkable), as well as adherens junctions (VE-Cadherin) and Nectin-2-containing junctions (*Figure 4*, *Figure 4—figure supplement 1*), similar as observed before (*Feldner et al., 2017*). In vitro experiments indicated that the co-localization of DLL1 and DLL4 with the adherens junction protein Nectin-2 at the cell membrane was diminished upon silencing *MPDZ* expression in ECs (*Figure 4—figure supplement 2A,B*), whereas silencing of Nectin-2 did not affect localization of DLL1 and DLL4 (*Figure 4—figure supplement 2C*). Taken together, the data suggest that MPDZ stabilizes the cell surface presentation of DLL1 and DLL4 through recruitment to the adherens junction protein Nectin-2.

## MPDZ regulates sprouting angiogenesis in vitro and ex vivo

DLL4/Notch signaling restricts sprouting angiogenesis and cells with high Notch signaling activity adopt the stalk cell phenotype in a growing vessel sprout (*Blanco and Gerhardt, 2013*; *Eilken and Adams, 2010*; *Pitulescu et al., 2017*). As such, the protein interaction of MPDZ with the Notch ligand DLL4 and its promotion of Notch signaling activity could potentially be important to control angiogenesis. To address this, *MPDZ* was silenced or over-expressed in HUVEC, which were embedded as spheroids into a collagen matrix to analyze endothelial sprout formation. HUVEC silenced for *MPDZ* expression had increased angiogenic potential and formed more capillary-like sprouts compared to control cells, both under basal conditions and after VEGF stimulation (*Figure 5A*). Oppositely, forced *MPDZ* expression resulted in impaired sprout formation under both conditions (*Figure 5B*). Furthermore, *MPDZ*-expressing cells, which exhibit increased Notch signaling activity, preferred the stalk cell position in the sprouting angiogenesis assay. In line with this result, *MPDZ* silenced cells, which showed less Notch signaling activity, adopted preferentially the tip cell position in a growing sprout (*Figure 5C*).

The increased angiogenic potential of *Mpdz*-deficient cells could also be shown in the aortic ring assay, in which slices of the mouse aorta from *Mpdz*$^{\Delta EC}$ or littermate control mice (*Mpdz*$^{fl/fl}$) were embedded into Matrigel. Aortic ECs from *Mpdz*$^{\Delta EC}$ mice showed a significantly higher outgrowth rate compared to control (*Figure 6A*).

## Loss of MPDZ promotes angiogenesis during brain development

Based on the in vitro and ex vivo data, we tested whether MPDZ also affects angiogenesis during mouse development. Vascularization of the hindbrain occurs in a stereotypic manner and is ideally suited to examine sprouting angiogenesis (*Fantin et al., 2013*). Hindbrains were resected at developmental stage E12.5 to analyze the vasculature. This revealed a much denser vessel network in *Mpdz*$^{-/-}$ embryos compared to wild-type littermates (*Figure 6B*). *Mpdz*$^{-/-}$ embryos had a significant higher number of vessel junctions and branches compared to littermate controls (*Figure 6C*). This indicates that Mpdz is needed to limit developmental angiogenesis.

## Endothelial-specific loss of Mpdz alters tumor angiogenesis

Angiogenesis is a hallmark of cancer and therefore we examined how genetic inactivation of *Mpdz* specifically in the endothelium would affect tumor growth and tumor angiogenesis. B16F10 melanoma and Lewis lung carcinoma (LLC) cells were injected subcutaneously into syngeneic *Mpdz*$^{\Delta EC}$

and control mice. No significant differences in the tumor growth rates were observed between *Mpdz*$^{fl/fl}$ and *Mpdz*$^{\Delta EC}$ mice (*Figure 7A and B*). The resected tumors were stained against the EC marker CD31 and α-SMA (smooth muscle cells, myofibroblasts) (*Figure 7C*). Microvessel density was significantly higher in *Mpdz*$^{\Delta EC}$ compared to *Mpdz*$^{fl/fl}$ mice (*Figure 7D*), whereas vessel coverage was not altered (*Figure 7D*), similar as observed after blocking Dll4-induced Notch signaling in the tumor vasculature (*Kangsamaksin et al., 2015*).

Increased numbers of blood vessels can support tumor growth, whereas too many vessel branches disturb the functionality of the vessel network, impair proper perfusion, inhibit tumor growth and can lead to tissue hypoxia. For instance, this was observed after inhibition of endothelial Dll4/Notch signaling in tumors (*Kangsamaksin et al., 2015*; *Noguera-Troise et al., 2006*; *Ridgway et al., 2006*). Indeed, both B16F10 and LLC tumors grown in *Mpdz*$^{\Delta EC}$ mice contained larger hypoxic areas compared to controls, as indicated by Glut1 expression (*Figure 7E and F*). To elucidate whether tumor perfusion is impaired, we injected Hoechst 33342 and FITC-labeled *Lycopersicon Esculentum* lectin into a tail vein and resected the tumors 5 min later. This revealed that B16F10 as well as LLC tumors were less well perfused in *Mpdz*$^{\Delta EC}$ mice compared to control littermates (*Figure 7—figure supplement 1A and B*). In the melanoma model, the percentage of Lectin-positive tumor blood vessels was reduced in *Mpdz*$^{\Delta EC}$ mice (*Figure 7—figure supplement 1C*), whereas in the LLC model, which contains a better structured vasculature than the melanoma model, the Hoechst dye was delivered to a lesser extend into the tumor mass in *Mpdz*$^{\Delta EC}$ mice compared to control littermates (*Figure 7—figure supplement 1D*).

## Discussion

Notch signaling is of utmost importance to control numerous cell differentiation steps during development. The activation of Notch receptors depends on physical contact with Notch ligands expressed on the adjacent cell. This study describes a novel mechanism that improves presentation of Delta-like ligands on the cell surface. MPDZ could be identified as a protein that mediates intracellular protein interactions with DLL1, DLL4 and Nectin-2 to facilitate presentation of DLL1 and DLL4 at the EC surface and to strengthen Notch signaling activity.

Upon posttranslational modifications, Notch ligands are transported to the plasma membrane. It is not yet clear how Notch ligands are recruited to the plasma membrane or to cellular junctions. Previous studies have shown that PDZ domain proteins interact with Notch ligands (*Adam et al., 2013*; *Ascano et al., 2003*; *Mizuhara et al., 2005*; *Pfister et al., 2003*; *Six et al., 2004*; *Wright et al., 2004*). However, the functional consequences of these are mostly elusive. The interaction of DLL1 with SYNJ2BP acts via a different mechanism on Notch signaling strength. SYNJ2BP prevents lysosomal degradation of DLL1 (*Adam et al., 2013*). The interaction with MPDZ however promotes cell surface presentation. Based on this, one can assume that changes in expression levels of Notch ligand-interacting PDZ proteins influence strongly the behavior of Notch ligands.

Here, we demonstrated that MPDZ interacts with Delta-like ligands and the transmembrane protein Nectin-2, a component of adherens junctions. This is interesting as also Notch receptors can be enriched at adherens junctions (*Batchuluun et al., 2017*; *Benhra et al., 2011*; *Hatakeyama et al., 2014*; *Sasaki et al., 2007*) to facilitate the physical interactions with ligands. Indeed, we found evidence that MPDZ mediates Notch signaling in cultured cells and isolated ECs derived from *Mpdz*-deficient mice. Whereas forced *MPDZ* expression enhanced Notch target gene expression, inactivation of the *MPDZ* gene resulted in a lower Notch target gene expression which might be due to the reduced DLL1 and DLL4 localization at the cell surface.

The reduction of endothelial Notch signaling activity in *Mpdz*$^{\Delta EC}$ mice was only moderate. One possible explanation is the redundancy between PDZ proteins that bind the same motif. For example, MPDZ shares large structural homology with the INADL protein (*Adachi et al., 2009*) and the Notch ligands can be bound also by several other PDZ domain proteins (*Adam et al., 2013*; *Estrach et al., 2007*; *Pfister et al., 2003*; *Six et al., 2004*; *Wright et al., 2004*). As such it is not surprising that *Mpdz*-deficient mice did not exhibit major angiogenesis defects, which would cause embryonic lethality. The vascular phenotype is similar to *Notch1* heterozygous mice. These mutants develop normally and show only slight vascular abnormalities (higher microvessel density and vessel branching) during phases of rapid vessel growth (*Huppert et al., 2000*).

Pharmacological Notch inhibition also leads to excessive tumor vessel sprouting and the formation of a poorly functional vessel network due to too many branches (*Funahashi et al., 2008*; *Noguera-Troise et al., 2006*; *Ridgway et al., 2006*). This is in particular achieved by DLL4-specific, but not JAG1-specific inhibition of Notch signaling (*Kangsamaksin et al., 2015*). We could also observe an increased tumor vessel sprouting in mice lacking *Mpdz* expression in the endothelium. Tumor vessel density was significantly increased, tumor perfusion impaired and this led to larger hypoxic tumor areas. Again, similar as after DLL4 blockade, vessel coverage with mural cells was not altered in tumors grown in *Mpdz*$^{\Delta EC}$ mice (*Funahashi et al., 2008*; *Kangsamaksin et al., 2015*). Taken together, this work shows that MPDZ is a novel modulator of DLL4-induced Notch signaling in the vasculature by recruiting DLL1 and DLL4 to Nectin-2 on the cell surface.

# Materials and methods

## Key resources table

| Reagent type (species) or resource | Designation | Source or reference | Identifiers | Additional information |
|---|---|---|---|---|
| Strain, strain background (*Mus musculus*) | Mpdz$^{-/-}$ | DOI: 10.15252/emmm. 201606430 | | |
| Strain, strain background (*Mus musculus*) | Mpdz$^{\Delta EC}$ | DOI: 10.15252/emmm. 201606430 | | |
| Genetic reagent (*Homo sapiens*) | MPDZ shRNA | Biocat | V2LHS_3656, 16945 16946 | |
| Genetic reagent (*Homo sapiens*) | Nectin-2 siRNA | Origene | SR321541 | |
| Antibody | MPDZ | Sigma-Aldrich | HPA020255 | Western Blot (1:500)/ICC (1:50) on methanol fixed cells |
| Antibody | Mpdz | Invitrogen | 42–2700 | Western Blot (1:500) |
| Antibody | HA | Cell Signaling | #3724 | Western Blot (1:1000) |
| Antibody | Flag | Sigma-Aldrich | F3165 | Western Blot (1:1000) |
| Antibody | DLL1 | abcam | ab85346 | Western Blot (1:1000)/ICC (1:100) on PFA fixed cells |
| Antibody | Dll1 | R and D Systems | AF3970 | Western Blot (1:500) |
| Antibody | DLL4 | Cell Signaling | #2589 | Western Blot (1:1000) |
| Antibody | DLL4 | Sigma-Aldrich | WH0054567M4 | ICC (1:100) on PFA fixed cells |
| Antibody | Dll4 | R and D Systems | AF1389 | Western Blot (1:500) |
| Antibody | SYNJ2BP | abcam | ab69431 | Western Blot (1:250) |
| Antibody | GFP | abcam | ab290 | Western Blot (1:2500) |
| Antibody | Nectin-2 | abcam | ab135246 | ICC (1:50) on PFA fixed cells |
| Antibody | Nectin-2 | Santa Cruz Biotechnology | sc-32804 | Western Blot (1:500)/ICC (1:50) on PFA fixed cells |
| Antibody | Nectin-2 | abcam | ab16912 | ICC (1:100) on fresh frozen sections |
| Recombinant DNA reagent | MPDZ | BioCat | clone BC140793 | |
| Recombinant DNA reagent | Citrine-MPDZ | this paper | | Citrine tag fused to the N-terminus of MPDZ (BioCat, clone BC140793) |
| Recombinant DNA reagent | DLL1 | OpenBiosystems | OHS4559-99847851 | |

*Continued on next page*

Continued

| Reagent type (species) or resource | Designation | Source or reference | Identifiers | Additional information |
|---|---|---|---|---|
| Recombinant DNA reagent | HA-DLL1 | this paper | | Gateway cloning: Dll1 (OHS4559-99847851) into pDest26-HA |
| Recombinant DNA reagent | DLL1-mCherry | this paper | | mCherry inserted between extracellular and transmembrane domain of DLL1. Gateway vector: pLenti6.2 |
| Recombinant DNA reagent | HA-DLL1ΔPDZ | this paper | | Stop codon inserted before PDZ-binding site by site-directed mutagenesis. Gateway vector: pDest26-HA |
| Recombinant DNA reagent | DLL1-mCherry ΔPDZ | this paper | | Stop codon inserted before PDZ-binding site by site-directed mutagenesis. Gateway vector: pLenti6.2 |
| Recombinant DNA reagent | DLL4 | PMID:17045587 | | cDNA cloned into pEntr3C |
| Recombinant DNA reagent | Flag-DLL4 | this paper | | cDNA cloned into pCS2p-FLAG |
| Recombinant DNA reagent | DLL4-mCherry | this paper | | mCherry inserted between extracellular and transmembrane domain of DLL4. Gateway vector: pLenti6.2 |
| Recombinant DNA reagent | Flag-DLL4ΔPDZ | this paper | | Stop codon inserted before PDZ-binding site by site-directed mutagenesis. Vector: pCS2p-FLAG |
| Recombinant DNA reagent | DLL4-mCherry ΔPDZ | this paper | | Stop codon inserted before PDZ-binding site by site-directed mutagenesis. Gateway vector: pLenti6.2 |
| Recombinant DNA reagent | Nectin-2 | DKFZ Genomics and Proteomics Core Facility | | BC003091 (cDNA). Gateway vector: pLenti6.2 |

## Animal experiments

Mice were kept under pathogen-free barrier conditions. All animal procedures were performed in accordance with the institutional and national regulations and approved by the local committees for animal experimentation and the local government (reference number: 35–9185.81/G-30/14 and 35–9185.81/G-259/12). Generation of global *Mpdz*$^{-/-}$ mice and conditional *Tek*-Cre;*Mpdz*$^{fl/fl}$ mice was previously described (*Feldner et al., 2017*). Mice had been backcrossed on a C57Bl/6 background for 10 generations. For tumor experiments, 500,000 syngeneic tumor cells (B16F10 or LLC) in 100 µl PBS were injected subcutaneously in the abdominal flanks of mice. To analyze tumor perfusion, 100 µl of fluorescein-labeled lectin (1 mg/ml, FL-1171, Vector Laboratories, Burlington, CA) and Hoechst 33342 (5 mg/ml, H3570, Life Technologies, Carlsbad, CA) was injected into the tail vein 5 min prior to sacrifice.

## Cell culture

B16F10, LLC and HEK293T cells were cultured in DMEM containing 10% fetal calf serum, 100 units/ml penicillin and 100 µg/ml streptomycin. Primary human umbilical cord endothelial cells (HUAEC and HUVEC) were freshly isolated and cultured in Endopan-3 medium with supplements (P04-0010K, PAN Biotech, Aidenbach, Germany) and used until passage five. HUVEC of three donors were

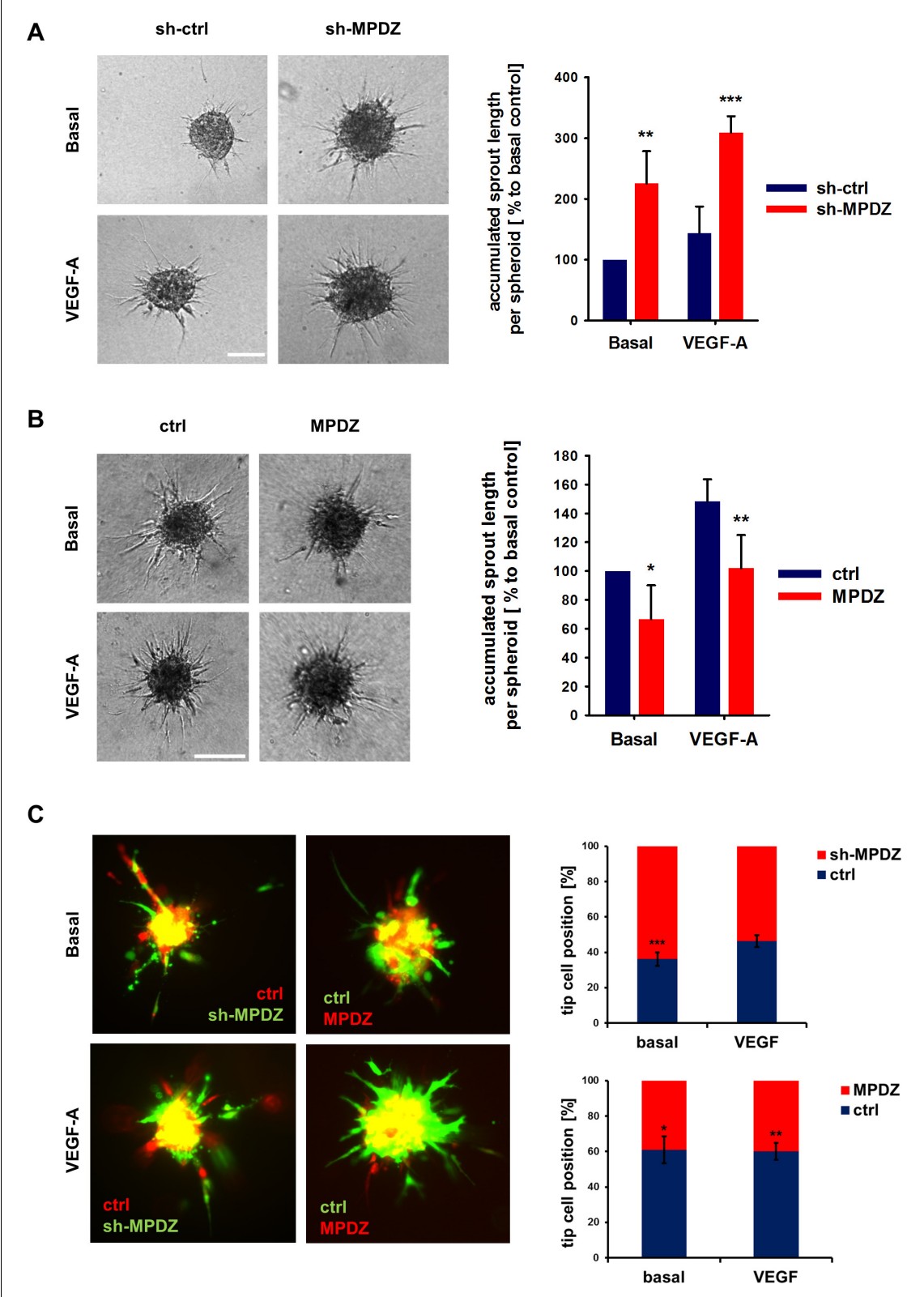

**Figure 5.** MPDZ inhibits sprouting angiogenesis in vitro. (**A**) HUVEC were transduced with lentivirus-expressing shRNA against MPDZ (sh-MPDZ) or expressing GFP (sh-ctrl). Sprouting angiogenesis of collagen-embedded spheroids was analyzed 72 hr after transduction. Spheroids were cultured under basal conditions or stimulated with VEGF-A (25 ng/ml). Quantification shows length of all sprouts of each spheroid. n = 4 experiments with 10 spheroids per condition. **, p<0.01; ***, p<0.001 one-way ANOVA (Holm-Sidak method). (**B**) HUVEC were transduced with control (ctrl) or MPDZ-
*Figure 5 continued on next page*

*Figure 5 continued*
expressing adenovirus. Sprouting angiogenesis of collagen-embedded spheroids was analyzed 72 hr after transduction. Spheroids were cultured under basal conditions or stimulated with VEGF-A (25 ng/ml). Quantification shows length of all sprouts of each spheroid. n = 5 experiments with 10 spheroids per condition. *, p<0.05; **, p<0.01; One Way ANOVA (Holm-Sidak method). (C) Mixed spheroids of HUVEC transduced with lentivirus expressing shRNA against MPDZ (sh-MPDZ) or expressing mCherry (ctrl) or spheroids of HUVEC transduced with control (ctrl) or MPDZ-expressing (MPDZ) adenovirus were embedded in a collagen matrix and analyzed 72 hr after transduction. Cells at the most distal end were considered as tip cells. Tip cell numbers were analyzed under basal conditions or after stimulation with VEGF-A (25 ng/ml). n = 3 experiments with 10 spheroids per condition. *, p<0.05; **, p<0.01; ***, p<0.001 unpaired Student's t-test.
DOI: https://doi.org/10.7554/eLife.32860.014
The following source data is available for figure 5:

**Source data 1.** Source data of the sprouting assay and the tip-stalk-cell competition assay related to *Figure 5 A, B, C*.
DOI: https://doi.org/10.7554/eLife.32860.015

pooled. Human brain microvascular endothelial cells were also cultured in Endopan-3 medium with supplements (P04-0010K, PAN Biotech). Standardized multiplex cell contamination and cell line authentication testing (Multiplexion, Heidelberg, Germany) were conducted on a regular basis.

## Expression plasmids, viral transduction, transfection

HUVECs were transduced with lentivirus and adenovirus as described (*Brütsch et al., 2010*). For MPDZ silencing three different lentiviral shRNA vectors (Biocat V2LHS_3656, 16945 16946) were used (*Feldner et al., 2017*). Forced MPDZ expression was achieved by lentiviral or adenoviral transduction expressing MPDZ cDNA (BioCat clone BC140793). Forced expression of Nectin-2 was achieved by transient over-expression of Nectin-2 cDNA in HEK293T cells (DKFZ clone BC003091). SYNJ2BP, DLL1 and DLL4 expression constructs were described (*Adam et al., 2013*; *Diez et al., 2007*). Mutations in cDNAs were introduced by site-directed mutagenesis using the QuickChange XL Kit (Agilent, Santa Clara, CA). Mutagenesis primers for the deletion of the DLL1-PDZ-binding site were as follows: 5'-gagaaggatgagtgcgtctaagcaactgaggtgtaagg-3', 5'-ccttacacctcagttgcttagacgcactcatccttctc-3'. Mutagenesis primers for the deletion of the DLL4-PDZ-binding site were as follows: 5'-gaggagaggaatgaatgtgtctatgccacggaggtataagg-3', 5'-ccttatacctccgtggcatagacacattcattcctctcctc-3', 5'-ggagaggaatgaatgtgtctaagccacggaggtat-3', 5'-atacctccgtggcttagacacattcattcctctcc-3'.

HUVECs were transfected with siRNA using RNAiMAX transfection reagent (Life Technologies). For Nectin-2 silencing, three different siRNAs were used (SR321541, Origene, Rockville, MD).

## Immunoprecipitation and western blot analysis

Cells were lysed with Cell Lysis Buffer (9803S, Cell Signaling Technology, Danvers, MA) complemented with 1 mM PMSF. Proteins were separated by SDS-PAGE and blotted on nitrocellulose membranes. Membranes were blocked with 5% skim milk in PBS containing 0.05% Tween-20. Following primary antibodies were used: anti-HA (#3724, Cell Signaling Technology, 1:1000), anti-FLAG (F3165, Sigma-Aldrich, St. Louis, MO, 1:1000), anti-DLL1 (ab85346, abcam, Cambridge, UK, 1:1000), anti-DLL4 (#2589S, Cell Signaling Technology, 1:1000), anti-beta-actin (A5441-.2ML, Sigma-Aldrich, 1:2500), anti-DLL1 (AF3970, R & D Systems, Minneapolis, MN, 1:500), anti-DLL4 (AF1389, R & D Systems, 1:250), anti-MPDZ (HPA020255, Sigma-Aldrich, 1:500), anti-Mpdz (42–2700, Invitrogen, Carlsbad, CA, 1:500), anti-GFP (ab290, abcam, 1:1000), anti-SYNJ2BP (ab69431, abcam, 1:500), anti-Nectin-2 (sc-32804, Santa Cruz Biotechnology, Dallas, TX, 1:500). Membranes were incubated overnight at 4°C with primary antibodies. HRP-conjugated secondary antibodies (DAKO, Santa Clara, CA) were added for 1 hr at room temperature. Chemoluminescence was detected by Aceglow ECL Western Blotting Substrate (PEQL37-3420, VWR International, Darmstadt, Germany) using a ChemiDoc imaging system (Bio-Rad Laboratories, Hercules, CA). Western Blots were quantified with Image Lab 3.0 software (Bio-Rad Laboratories).

For immunoprecipitation, primary antibodies were added to protein lysates and incubated overnight at 4°C. Protein-G-coupled dynabeads (10003D, Invitrogen) were added to the protein lysates at the next day and were incubated for 30 min at 4°C. Precipitated dynabeads were washed three times with ice-cold PBS and denatured in Laemmli sample buffer at 95°C for 5 min. Samples were then subjected to Western blotting.

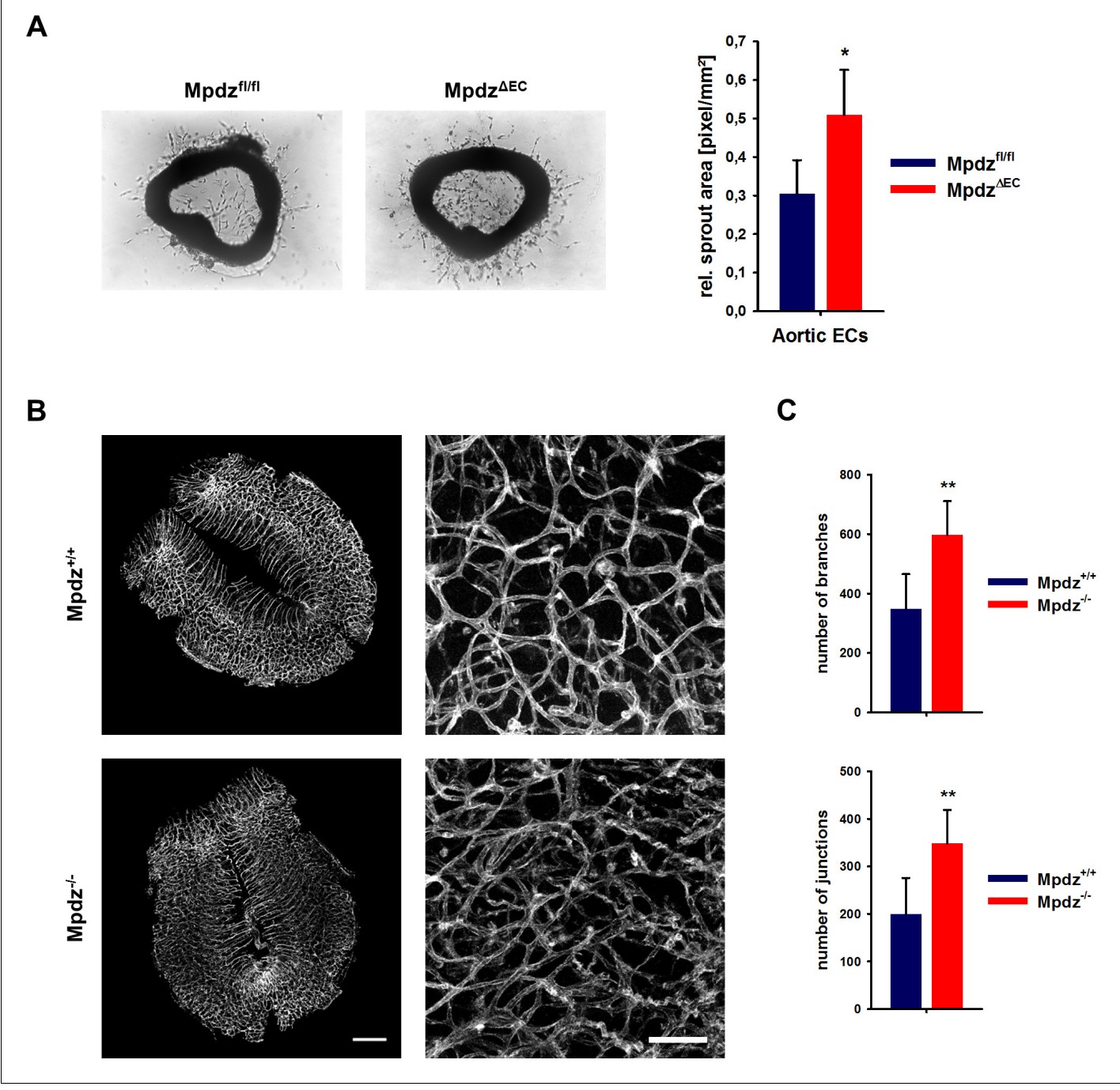

**Figure 6.** Loss of Mpdz leads to increased vessel branching in the embryonic mouse hindbrain. (**A**) Aortae were isolated from *Mpdz*^fl/fl and *Mpdz*^ΔEC mice. Aortic rings were embedded in Matrigel and EC outgrowth was analyzed 24 hr after embedding. n = 4 mice per genotype; *, p<0.05 unpaired Student's t-test. (**B**) Embryos at developmental stage E12.5 were used for IsolectinB4-FITC staining (endothelial cells). Left panel shows whole hindbrains of *Mpdz*^+/+ and *Mpdz*^-/- embryos. Right panel shows zoom-ins. Left panel: scale bar, 500 μm; Right panel: scale bar, 100 μm. (**C**) Quantification of vessel branches and junctions per area. n ≥ 6 mice per genotype; **, p<0.01 unpaired Student's t-test.

DOI: https://doi.org/10.7554/eLife.32860.016

The following source data is available for figure 6:

**Source data 1.** Source data of the aortic ring assay and the blood vessel analysis of embryonic hindbrains related to *Figure 6 A and C*.
DOI: https://doi.org/10.7554/eLife.32860.017

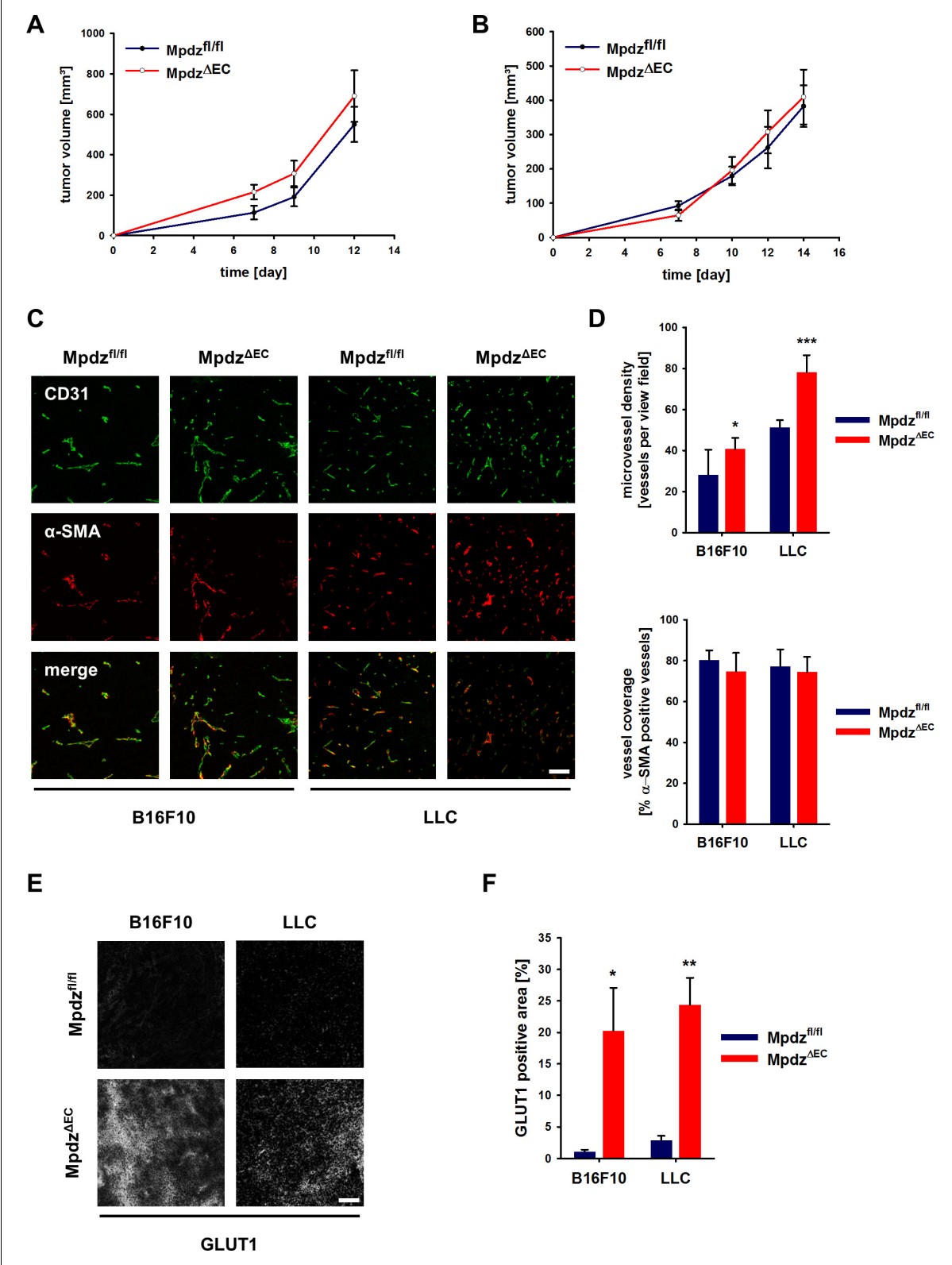

**Figure 7.** Excessive tumor angiogenesis upon endothelial-specific inactivation of Mpdz. Tumor growth curve of B16F10 (A) and LLC (B) tumors subcutaneously implanted into *Mpdz*^fl/fl and *Mpdz*^ΔEC mice. n ≥ 5; Results are shown as mean ±SEM. (C) Representative images of B16F10 and LLC tumors grown in *Mpdz*^fl/fl and *Mpdz*^ΔEC mice, stained against CD31 (endothelial cells) and α-SMA (smooth muscle cells). Scale bar: 100 μm (D) Quantification of the vessel staining. Microvessel density was determined by counting the CD31-positive vessels per area. For the analysis of vessel

*Figure 7 continued on next page*

*Figure 7 continued*

coverage, the percentage of α-SMA-positive vessels was determined. n ≥ 5; results are shown as mean ±SD; *, p<0.05; ***, p<0.001; unpaired Student's t-test. (E) Representative images of B16F10 and LLC tumors stained for Glut1 (hypoxia marker) grown in *Mpdz*^fl/fl and *Mpdz*^ΔEC mice. Scale bar: 100 μm (F) Quantification of the Glut1-positive area. n ≥ 4; results are shown as mean ±SEM; *, p<0.05; **, p<0.01; unpaired Student's t-test. Figure legends – figure supplements.

DOI: https://doi.org/10.7554/eLife.32860.018

The following source data and figure supplement are available for figure 7:

**Source data 1.** Source data of the microvessel density analysis and the Glut1 expression analysis related to *Figure 7D and F*.
DOI: https://doi.org/10.7554/eLife.32860.020

**Figure supplement 1.** Excessive tumor angiogenesis upon endothelial-specific inactivation of Mpdz.
DOI: https://doi.org/10.7554/eLife.32860.019

Quantitative Real-Time-PCR mRNA was isolated with the innuPrep RNA Mini Kit (845-KS-2040250, Jena Analytics, Jena, Germany) and transcribed into cDNA (High Capacity cDNA Reverse Transcription Kit; 4368814, Applied Biosystems, Foster City, CA). POWER SYBR Green Master Mix (4368708, Applied Biosystems) was used to perform qPCR on an ABI StepOnePlus cycler (Applied Biosystems). *Rpl32* and *OAZ1* were used for normalization. Primers: h*OAZ* (fw): 5'-gagccgaccatgtctt-catt-3', h*OAZ* (rev): 5'-ctcctcctctccccgaagact-3'; m*Rpl32* (fw): 5'-aggcattgacaacagggttc-3', m*Rpl32* (fw): 5'-gttgcacatcagcagcactt-3'; h*HEY1* (fw): 5'-gagaaggctggtacccagtg-3', h*HEY1* (rev): 5'-cgaaatcc-caaactccgata-3'; m*Hey1* (fw): 5'-gaaaagacggagaggcatca-3', m*Hey1* (rev): 5'-gtgcgcgtcaaaataacctt-3'; h*HEY2* (fw): 5'-cttgtgccaactgcttttga-3', h*HEY2* (rev): 5'-gcactctcggaatcctatgc-3'; m*Hey2* (fw): 5'- tga-gaagactagtgccaacagc-3', m*Hey2* (rev): 5'-tgggcatcaaagtagcctta-3'; h*HES1* (fw): 5'-tcaacacgacaccg-gataaa-3', h*HES1* (rev): 5'-ccgcgagctatctttcttca-3'. All experiments included two technical and three biological replicates.

## Luciferase co-culture assay

A Notch expressing cell line, CHO-N1-CIT, was transfected in 24-well dishes using TransIT-LT1 (Mirus Bio, Madison, WI) with a TP1-firefly Notch luciferase reporter (800 ng) together with SV-40 Renilla luciferase (10 ng) (*Shaya et al., 2017*). IMCD3 cells were transfected under similar conditions with either an MPDZ construct or empty vector (pORI) as control. 24 hr after transfection, cells were tryp-sinized and the CHO-N1-CIT cells were co-cultured with either IMCD3 cells transfected with MPDZ or pORI. The cells were plated in a ratio of 40:60 (IMCD3: CHO-N1-CIT) and were incubated for 48 hr, after which the cells were lysed with lysis buffer (E1960, Promega, Madison, WI). The light emis-sion of the luciferin and the Renilla luciferase activities were measured from cell lysates using a dual luciferase kit (E1960, Promega) and a Veritas luminometer (Promega). The assay was repeated five independent times.

## Spheroid-based sprouting angiogenesis

HUVECs were suspended in growth medium containing 20% methocel (Sigma-Aldrich). Endothelial cells were cultured as hanging drops for 24 hr to form spheroids. Each spheroid contained approxi-mately 400 cells. Spheroids were suspended in 2 ml methocel containing 20% FCS and 2 ml rat colla-gen at neutral pH. The collagen matrix polymerized for 30 min and hereon 0.1 ml basal culture medium or 0.1 ml basal culture medium containing VEGF-A (final concentration 25 ng/ml; Pepro-tech, Hamburg, Germany) was added. After 24 hr, cells were fixed with 10% formaldehyde. The lengths of all sprouts of at least 10 spheroids per condition were measured using an inverted micro-scope (Leica DM IRB, Leica Microsystems, Wetzlar, Germany). Image analysis was done by using Fiji software. The assay is described in more detail at Bio-protocol (*Tetzlaff and Fischer, 2018*).

To determine which cells prefer the tip or stalk cell position, a sprouting assay with a co-culture of endothelial cells was performed. Cells expressed either a fluorophore (GFP or mCherry) or were labeled with Cell Tracker Red (C34552, Life Technologies). Equal amount of each cell type were cul-tured as hanging drops. Sprouting assay was performed as described above. Using the inverted microscope (Leica DM IRB), the number of green or red fluorescent cells in the tip cell position was determined. Per condition at least 10 spheroids were analyzed.

## Hindbrain analysis

Embryonic hindbrains were isolated as described previously (*Fantin et al., 2013*). Samples were fixed with 4% PFA overnight at 4°C and hereon permeabilized in blocking buffer (0.3% Triton X-100% and 1% BSA in PBS) overnight at 4°C. Samples were washed three times for 20 min with Pblec buffer at room temperature and stained with FITC-IsolectinB4 (1:100; L2895, Sigma-Aldrich) in Pblec overnight at 4°C. After washing, samples were mounted using fluorescence mounting medium (S3023, DAKO). Z-stack images were acquired using a confocal microscope (Zeiss LSM 700, Zeiss, Oberkochen, Germany) and image analysis was done by Fiji software.

## Aortic ring assay

Aortae were isolated from mice (8 weeks old) and cut into ~25 rings each. Aortic rings were embedded in matrigel (356234, BD Biosciences, Franklin Lakes, NJ), and stimulated with 30 ng/ml VEGF-A$_{165}$(450–32, Peprotech). Images were taken after 24 hr with a Nikon SMZ800 microscope.

## Immunofluorescence

Freshly dissected tumors were embedded in Tissue-Tek (4583, Sakura), frozen and stored at −80°C. Sections (7 μm) were cut and fixed in methanol for 20 min at −20°C. The primary antibodies against CD31 (550274, BD Biosciences, 1:50), α-SMA (C6198, Sigma-Aldrich, 1:100) and Glut1 (ab40084, abcam, 1:200) were incubated over night at 4°C and secondary antibodies (Thermo Fisher Scientific, 1:400) for 1 hr at room temperature. Sections were washed three times with TBS-T and mounted with Fluoromount (S3023, Dako). Confocal images were obtained using an LSM 700 microscope (Zeiss) and analyzed using Fiji software.

Retinae were isolated, fixed and processed as previously described (*Yang et al., 2015*). Specimens were stained for IsolectinB4 (1:100; L2895, Sigma-Aldrich), Claudin 5 (1:100, ab53765, abcam), VE-Cadherin (1:100, 555289, BD Biosciences), Nectin-2 (1:50, ab16912, abcam).

HUVEC were seeded on glass slides coated with 0.5% gelantine. Cells were washed twice with PBS, fixed with 4% PFA for 10 min, washed three times with PBS, permeabilized with PBS-T (containing 0.1% TritonX) and washed again three times with PBS. Alternative to PFA fixation, cells were fixed with ice-cold methanol for 20 min at −20°C and then washed three times with PBS. After blocking with 3% BSA in PBS, cells were incubated with the primary antibodies against MPDZ (HPA020255, Sigma-Aldrich, 1:50), DLL1 (ab85346, abcam, 1:100), DLL4 (WH0054567M4, Sigma-Aldrich, 1:100), Nectin-2 (ab135246, abcam, 1:50), Nectin-2 (sc-32804, Santa Cruz Biotechnology, 1:50) over night at 4°C and secondary antibodies (Thermo Fisher Scientific, 1:400) for 1 hr at room temperature. Sections were counterstained with DAPI, washed three times with PBS and mounted with Fluoromount (S3023, Dako). Confocal images were obtained using an LSM 700 microscope (Zeiss) and analyzed using Fiji software.

## Flow cytometry

HUVECs were detached from cell culture plates using trypsin-EDTA (25300054, Thermo Fisher Scientific, Waltham, MA). Mouse embryos were minced and incubated with 0.5 mg/ml collagenase type II (LS004177, Worthington, Lakewood, CA) for 45 min at 37°C. Tissue suspensions were mashed twice through cell strainers (BD Biosciences; 100 and 40 μm). Endothelial cells were enriched by CD31 magnetic beads. Cells were suspended ($10^6$ cells/ml) and incubated with different fluorophores coupled to primary antibodies against DLL1 (FAB1818A, R & D Systems), DLL4 (FAB1506A, R & D Systems), Dll4 (563802, BD Biosciences), CD34 (553733, BD Biosciences) for 20 min on ice. Concentration of the different antibodies was determined by titration, in order to get optimal compensation during acquisition.

## Statistical analysis

Statistical analysis was performed with SigmaPlot software 12.5 (Systat Software Inc., San Jose, CA). Statistical significance was calculated using unpaired Student's t-test and one-way ANOVA, as adequate. p-values<0.05 were considered as significant.

## Study approval

All animal works were approved by the local committees for animal experimentation and the local government (reference number: 35–9185.81/G-30/14 and 35–9185.81/G-259/12). This work is not considered 'Human Subjects Research'.

## Acknowledgements

We thank Christian Clappier, Fabienne Podieh and Maaike Schilperoort for excellent technical support, Damir Krunic (DKFZ, light microscopy core facility) for help with FIJI software data analysis, Bernd Heßling (DKFZ, genomics and proteomics core facility) and members of the DKFZ Laboratory Animal Facility for support.

## Additional information

### Funding

| Funder | Grant reference number | Author |
| --- | --- | --- |
| Deutsche Forschungsge-meinschaft | SFB-TR23 (A7) | Andreas Fischer |
| Deutsches Krebsforschungs-zentrum and the Israeli Ministry of Science | Cooperation Program in Cancer Research CA156 | Fabian Tetzlaff Amitai Menuchin David Sprinzak Andreas Fischer |
| Helmholtz-Gemeinschaft | | Andreas Fischer |

The funders had no role in study design, data collection and interpretation, or the decision to submit the work for publication.

### Author contributions

Fabian Tetzlaff, Conceptualization, Data curation, Formal analysis, Validation, Investigation, Visualization, Methodology, Writing—original draft; M Gordian Adam, Conceptualization, Data curation, Formal analysis, Validation, Investigation, Methodology, Writing—review and editing; Anja Feldner, Conceptualization, Data curation, Formal analysis, Validation, Investigation, Methodology; Iris Moll, Amitai Menuchin, Data curation, Formal analysis, Validation, Investigation, Methodology; Juan Rodriguez-Vita, Supervision, Writing—review and editing; David Sprinzak, Resources, Funding acquisition, Writing—review and editing; Andreas Fischer, Conceptualization, Resources, Formal analysis, Supervision, Funding acquisition, Writing—original draft, Project administration

### Author ORCIDs

Juan Rodriguez-Vita (iD) http://orcid.org/0000-0001-9547-5508
David Sprinzak (iD) http://orcid.org/0000-0001-6776-6957
Andreas Fischer (iD) http://orcid.org/0000-0002-4889-0909

### Ethics

Animal experimentation: Mice were kept under pathogen-free barrier conditions. All animal procedures were performed in accordance with the institutional and national regulations and approved by the local committees for animal experimentation (Heidelberg University and DKFZ) and the local government (Regierungspräsidium Karlsruhe, Germany).(reference number: 35-9185.81/G-30/14 and 35-9185.81/G-259/12).

### Decision letter and Author response

Decision letter https://doi.org/10.7554/eLife.32860.024
Author response https://doi.org/10.7554/eLife.32860.025

## Additional files

**Supplementary files**
• Transparent reporting form
DOI: https://doi.org/10.7554/eLife.32860.021

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
