## [Decision Letter]

Thank you for submitting your article "MPDZ promotes DLL4-induced Notch signaling during angiogenesis" for consideration by *eLife*. Your article has been favorably evaluated by Didier Stainier (Senior Editor) and three reviewers, one of whom, Elisabetta Dejana, is a member of our Board of Reviewing Editors. The following individual involved in review of your submission has agreed to reveal their identity: Lena Claesson-Welsh. The Reviewing Editor has drafted this decision to help you prepare a revised submission.

As you will see from their comments both reviewers liked your work and acknowledge the novelty and the relevance of your observations. However, both reviewers ask for some additional experimental data that, to our view, would strongly improve the quality and the strength of the paper. After discussion among the reviewers, we ask you to consider the specific recommendation #1 of reviewer 1 and #4 of reviewer 2 as essential. We ask you to consider the other remarks but view them as advisory rather than essential.

Reviewer #1:

In this paper the authors present novel data on the Notch signaling system and cell-to-cell junctions. In particular, they found that MPDZ (multiple PDZ protein) associates to Dll1 and Dll4 promoting their interaction with Nectin-2 and modulating signaling. Although of interest, the paper presents some weak experimental aspects that do not fully support the conclusions.

More specifically:

1) The authors should show immunofluorescence junctional staining of endothelial cells with Dll1 and Dll4 in the presence or absence of MPDZ and in the presence or absence of Nectin-2, in sparse and confluent conditions. This is required to support immuno- precipitation data and to exclude any possible in vitro association of MPDZ, nectin and Dll1 and 4.

These experiments would be even more convincing by performing the analysis of junction organization and composition in vivo in MPDZ ECKO mice.

2) The results presented do not exclude that Nectin-2 could associate to Dll4 also in absence of MPDZ (see Figure 3E using 293 T cells). The amount of DLL4 co-immunoprecipitated with Nectin-2 in absence of MPDZ is lower but still easily detectable.

3) Along the paper the authors use a quite large spectrum of different cell lines (HEK293T, HUVEC, cardiac and lung EC). It is conceivable that each cell line may express other proteins able to modulate Notch signaling. I fully understand the difficulties in transfecting different types of endothelial cells but, for sake of clarity, I would make an effort to maintain the same endothelial cell line along the work.

Reviewer #2:

The study by Tetzlaff et al., follows up on a yeast-two hybrid screen for Dll interacting proteins in which the authors identified the multiple PDZ domain protein (MPDZ). They now verify this interaction in cell models and show that MPDZ gain-/loss of function leads to increased/decreased expression of HEY2 and HES1 indicating that MPDZ modulates Notch signalling activity. At least in part, this effect is exerted by regulation of the subcellular localization of Dll ligands as their complex formation with the adherens junction protein Nectin-2 is dependent on MPDZ. The authors probe for the effect of MPDZ on vessel formation using in vitro (sprouting collagen assay), ex vivo (aortic rings with and without siRNA silencing) and in vivo models, developmental (hindbrain) as well as pathological (cancer). The data are neatly presented, well controlled and of interest to the vascular biology field.

1) In a yeast-two hybrid screen for proteins interacting with the intracellular domains of Dll ligands, the authors previously identified SYNJ2BP and MUPP1/MPDZ. Forced lentiviral SYNJ2BP expression led to increased transcription of the HEY1 and LFNG, indicating increased Notch activity (Adam et al., Circ Res 2013) similar to the data in this study for MPDZ. What is the relationship between SYNJ2BP and MPDZ? Are they part of the same multiprotein complex? Do they compete for binding to Dll ligands? Which other proteins does MPDZ interact with in endothelial cells? It is likely that such multiprotein complexes are established at junctions and it is important to address these complexes in a comprehensive manner to understand their biology.

2) MPDZ has been described as a tight junction component in epithelial cells and as a Claudin-5 partner in Schwann cells, alternatively, as a gap junction protein. Please show the expression pattern of MPDZ in endothelial cells. Which types of vessels express MPDZ (arteries, veins, capillaries) and what is its subcellular localization?

3) Does MPDZ interact with Dll ligands in vivo? Can the authors use e.g. tumor lysates to ip for one and blot or do mass spectrometry to detect the other component?

4) The authors show that loss of MPDZ expression in endothelial cells leads to an increase in sprouting angiogenesis ex vivo (aortic rings, Figure 4) and during development (hindbrain angiogenesis, Figure 5). In cancer, the microvessel density is increased in the absence of MPDZ (Figure 6). Loss of MPDZ leads to higher Notch signalling and therefore increased angiogenesis which very nicely explains the findings in Figures 4-6. However, in none of these data sets are we informed about the morphology of the vasculature in the *Mpdz*-deficient condition compared to wt. Are MPDZ expressing ECs preferentially found at the tip cell position? Are the MPDZ LOF ECs tip cells? The tumors lacking MPDZ do not grow bigger but has a substantial hypoxic zone that is lacking in the wild type condition. Are the abundant vessels formed due to MPDZ-deficiency not functional? What is the phenotype of the tumor vasculature in Figure 6?

---

## [Author Response]

Reviewer #1:In this paper the authors present novel data on the Notch signaling system and cell-to-cell junctions. In particular, they found that MPDZ (multiple PDZ protein) associates to Dll1 and Dll4 promoting their interaction with Nectin-2 and modulating signaling. Although of interest, the paper presents some weak experimental aspects that do not fully support the conclusions.More specifically:1) The authors should show immunofluorescence junctional staining of endothelial cells with Dll1 and Dll4 in the presence or absence of MPDZ and in the presence or absence of Nectin-2, in sparse and confluent conditions. This is required to support immuno- precipitation data and to exclude any possible in vitro association of MPDZ, nectin and Dll1 and 4.These experiments would be even more convincing by performing the analysis of junction organization and composition in vivo in MPDZ ECKO mice.

We have addressed this point in several ways. First, we silenced MPDZ in HUVEC with lentiviral shRNA expression constructs and cultured the cells under sparse and confluent conditions. Cells were stained to detect DLL1, DLL4 and Nectin-2. Further staining against MPDZ was not possible, as this requires fixation in methanol whereas the other antibodies require PFA fixation. Only GFP positive cells were analyzed (indicating that they are expressing the transduced construct). Staining analysis revealed that the knock-down of MPDZ (verified by qPCR, WB and by the fact that lentiviral transduced cells express GFP) resulted in lesser co-localization of Nectin-2 with DLL1 or DLL4 (see Figure 4—figure supplement 2). Under confluent conditions, in particular the co-localization at the cell membrane was disturbed upon MPDZ silencing.

Next, we analyzed the localisation of Nectin-2 and DLL1/4 in HUVEC. Since Nectin-2 was transiently knocked-down, analysis under fully confluent conditions was not possible. We detected co-localization at the cell membrane. Silencing of Nectin-2 revealed that DLL1 and DLL4 were still at the same localization (see Figure 4—figure supplement 2). As such, MPDZ promotes the co-localization of DLL1/4 with Nectin-2, whereas Nectin-2 is not absolutely required for cell surface presentation of Notch ligands.

Secondly, we analyzed the junctional organization in retinal vessels of *Mpdz*^-/-^ and *Mpdz*^+/+^ mice (see new Figure 4) as well as in cryosections of brain derived from *Mpdz*^∆EC^ and *Mpdz*^fl/fl^ mice (see Figure 4—figure supplement 1). This revealed that loss of *Mpdz* does not affect the overall junctional organization and composition, what is not surprising as the mice are viable and do not show signs of edema. These data are also in line with our previous publication (Feldner et al., 2017).

2) The results presented do not exclude that Nectin-2 could associate to Dll4 also in absence of MPDZ (see Figure 3E using 293 T cells). The amount of DLL4 co-immunoprecipitated with Nectin-2 in absence of MPDZ is lower but still easily detectable.

We agree. However, this is difficult to address as the function of MPDZ can be compensated by INADL which shows high homology with MPDZ and which most likely interacts with the same proteins. For instance it has been shown that INADL also binds to Nectin-2 (Adachi et al., 2009). As such it might be possible that INADL compensates the loss of MPDZ. To investigate this in detail, we are currently generating *Mpdz* and *Inadl* double knock-out mice. Detailed analysis will be time consuming and goes beyond the scope of this manuscript.

3) Along the paper the authors use a quite large spectrum of different cell lines (HEK293T, HUVEC, cardiac and lung EC). It is conceivable that each cell line may express other proteins able to modulate Notch signaling. I fully understand the difficulties in transfecting different types of endothelial cells but, for sake of clarity, I would make an effort to maintain the same endothelial cell line along the work.

We fully agree that it would be ideal to perform all experiments in a single cell line. Therefore, we used HUVEC cells for almost all in vitro experiments. Only for biochemical studies we used HEK293T cells, since expression of cDNA constructs is more efficient. Nevertheless we wanted to confirm critical data also in ex vivo settings. Therefore we isolated also cardiac or lung EC from the KO mice. Here the isolation of two different EC types had at least the advantage of being able to perform mRNA expression (cardiac EC) together with protein expression (lung EC) in the same animal.

Reviewer #2:[…] 1) In a yeast-two hybrid screen for proteins interacting with the intracellular domains of Dll ligands, the authors previously identified SYNJ2BP and MUPP1/MPDZ. Forced lentiviral SYNJ2BP expression led to increased transcription of the HEY1 and LFNG, indicating increased Notch activity (Adam et al., Circ Res 2013) similar to the data in this study for MPDZ. What is the relationship between SYNJ2BP and MPDZ? Are they part of the same multiprotein complex? Do they compete for binding to Dll ligands? Which other proteins does MPDZ interact with in endothelial cells? It is likely that such multiprotein complexes are established at junctions and it is important to address these complexes in a comprehensive manner to understand their biology.

SYNJ2BP as well as MPDZ are PDZ-domain proteins which bind to DLL1 and DLL4 via their PDZ-binding site. To investigate whether MPDZ and SYNJ2BP compete for binding to the Notch ligands, we performed a Co-IP study with HEK293T control and *MPDZ*-silenced HEK293T cells (see Figure 1—figure supplement 1). DLL1 or DLL4 and SYNJ2BP were over-expressed and the Notch ligands were immunoprecipitated. Immunoblot analysis revealed that the absence of MPDZ does not affect the interaction of SYNJ2BP with the Notch ligands. Hence, we concluded that MPDZ and SYNJ2BP do not compete for binding to the Notch ligands. We fully agree with the reviewer that there will be most likely competition of PDZ proteins for binding to DLL ligands. However, at the moment we do not have the tools to study this in detail. We had reported before that SYNJ2BP prevents DLL1 lysosomal degradation. Here we propose that MPDZ promotes its localization at the cell membrane. Therefore it could indeed be that these binding proteins act at different subcellular localization. Unfortunately, the quality of the antibodies preclude any further investigation at this stage.

To address the question which proteins are also interacting with MPDZ in endothelial cells, we performed a mass spectrometry screen. MPDZ was immunoprecipitated in HUVEC lysates and the washed precipitate was analyzed by mass spectrometry (see Author response image 1). With this method we could show that the antibody indeed binds MPDZ and that we can verify the interaction with MPP5 (String database). However, in this screening approach we could not detect any of the known interactions with transmembrane proteins (e.g. Claudin-5, JAM-A or DLL4). The method of cell lysis or the composition of the precipitation buffer might be responsible for this. Therefore we suggest not to put this data set into the manuscript.

2) MPDZ has been described as a tight junction component in epithelial cells and as a Claudin-5 partner in Schwann cells, alternatively, as a gap junction protein. Please show the expression pattern of MPDZ in endothelial cells. Which types of vessels express MPDZ (arteries, veins, capillaries) and what is its subcellular localization?

We could show that MPDZ is expressed in endothelial cells derived from veins, arteries and microvessels (see Figure 3—figure supplement 1). We studied the subcellular localization in freshly isolated venous and arterial ECs (cell passage 0). This demonstrated that MPDZ is mainly expressed at the cell membrane but can be also found in the cytoplasm. In microvessel ECs we could not analyze the subcellular localization, since we had to use a commercial available cell line at a higher cell passage number (cell passage 4). Our previous experiments have clearly shown that MPDZ can only be detected robustly at the cell membrane in freshly isolated ECs (cell passage 0), ideally under confluent conditions.

3) Does MPDZ interact with Dll ligands in vivo? Can the authors use e.g. tumor lysates to ip for one and blot or do mass spectrometry to detect the other component?

To address this question we prepared total protein lysates from kidneys, which express high amounts of MPDZ, which is required for barrier integrity under high osmotic stress (Lanaspa et al., 2007). We immunoprecipitated DLL1 and DLL4 and immunoblotted for MPDZ. This revealed that MPDZ also interacted with DLL1 and DLL4 in this mouse tissue (Figure 1—figure supplement 1).

4) The authors show that loss of MPDZ expression in endothelial cells leads to an increase in sprouting angiogenesis ex vivo (aortic rings, Figure 4) and during development (hindbrain angiogenesis, Figure 5). In cancer, the microvessel density is increased in the absence of MPDZ (Figure 6). Loss of MPDZ leads to higher Notch signalling and therefore increased angiogenesis which very nicely explains the findings in Figures 4-6. However, in none of these data sets are we informed about the morphology of the vasculature in the Mpdz-deficient condition compared to wt. Are MPDZ expressing ECs preferentially found at the tip cell position? Are the MPDZ LOF ECs tip cells? The tumors lacking MPDZ do not grow bigger but has a substantial hypoxic zone that is lacking in the wild type condition. Are the abundant vessels formed due to MPDZ-deficiency not functional? What is the phenotype of the tumor vasculature in Figure 6?

To address the question whether MPDZ expressing cells prefer the tip or stalk cell position, we performed an in vitro tip-stalk-cell sprouting assay (new Figure 5C). *MPDZ*-deficient endothelial cells preferred the tip cell position whereas *MPDZ*-expressing cells adopted the stalk cell phenotype. This result is in accordance with a recent paper by the Adams laboratory showing that cells with high Notch signaling activity preferentially adopt the stalk cell phenotype (Pitulescu et al., 2017).

To further investigate the functionality of the tumor vessels, we repeated the tumor experiments and injected Hoechst 33342 and FITC-Lectin into the tail vein 5 min prior to sacrifice. Tumors were stained for CD31 and vessel perfusion and Hoechst extravasation was analyzed (Figure 7—figure supplement 1). The experiments in the wildtype control animals revealed that LLC tumors were better perfused than B16F10 tumors. In LLC tumors almost all of the CD31+ vessels were also Lectin positive. This was also the case in *Mpdz*^∆EC^ mice. However the rate of Hoechst dye penetration into the tumor mass was significantly reduced in *Mpdz*^∆EC^ mice compared to *Mpdz*^fl/fl^ control littermates. In B16F10 melanoma the Hoechst staining was very weak, which precluded further analysis. However, analysis of the Lectin-positive vessels revealed lower rates in *Mpdz*^∆EC^ mice compared to *Mpdz*^fl/fl^ littermates.

References:

Lanaspa, MA, Almeida, NE, Andres-Hernando, A, Rivard, CJ, Capasso, JM, and Berl, T. (2007). The tight junction protein, MUPP1, is up-regulated by hypertonicity and is important in the osmotic stress response in kidney cells. Proc. Natl. Acad. Sci. U. S. A. 104, 13672–13677.